# Type 2 Diabetes and Dietary Carbohydrate Intake of Adolescents and Young Adults: What Is the Impact of Different Choices?

**DOI:** 10.3390/nu13103344

**Published:** 2021-09-24

**Authors:** Luisa Bonsembiante, Giovanni Targher, Claudio Maffeis

**Affiliations:** 1Section of Pediatric Diabetes and Metabolic Disorders Unit, Department of Surgical Sciences, Dentistry, Paediatrics and Gynaecology, Azienda Ospedaliera Universitaria Integrata of Verona, Piazzale A. Stefani, 1, 37126 Verona, Italy; luisa.bonsembiante@studenti.univr.it; 2Section of Endocrinology, Diabetes and Metabolism, Department of Medicine, University and Azienda Ospedaliera Universitaria Integrata of Verona, Piazzale A. Stefani, 1, 37126 Verona, Italy; giovanni.targher@univr.it

**Keywords:** carbohydrates, nutrition, type 2 diabetes mellitus, adolescents, glycemic index

## Abstract

Type 2 diabetes mellitus has a high prevalence worldwide, with a rapidly increasing incidence even in youth. Nutrition, dietary macronutrient composition, and in particular dietary carbohydrates play a major role in the development of type 2 diabetes. The aim of this narrative review is to discuss the current evidence on the role of dietary carbohydrates in the prevention and management of type 2 diabetes. The digestibility or availability of carbohydrates and their glycemic index (and glycemic load) markedly influence the glycemic response. High consumption of dietary fiber is beneficial for management of type 2 diabetes, whereas high consumption of both glycemic starch and sugars may have a harmful effect on glucose metabolism, thereby increasing the risk of developing type 2 diabetes in the presence of genetic predisposition or making its glycemic control more difficult to achieve in people with established T2D. Therefore, the same dietary macronutrient may have harmful or beneficial effects on type 2 diabetes mainly depending on the subtypes consumed. Some other factors are involved in glucose metabolism, such as meal composition, gut microbiota and genetics. For this reason, the glycemic response after carbohydrate consumption is not easy to predict in the single individual. Nutrition suggested to subjects with known type 2 diabetes should be always person-centered, considering the individual features of each subject.

## 1. Introduction

Diabetes affects approximately 463 million people worldwide, and up to nearly 95% have type 2 diabetes mellitus (T2D) [1,2]. The prevalence of T2D has increased around fourfold in the last forty years [3,4]. Moreover, the age at T2D onset is progressively decreasing, and T2D is now also becoming common in youth [5]. Several factors may be involved in the anticipation of T2D onset, although the dramatic increase in childhood and adolescent obesity worldwide plays a key role in this process [5]. Similarly, the condition of prediabetes in youth has risen steeply worldwide in parallel with rising rates of obesity [6]. In the National Health and Nutrition Examination Survey cohort database, 13.1% of United States adolescents had impaired fasting glycemia, and 3.4% had impaired glucose tolerance after oral glucose load. In addition, the prevalence of prediabetes was even greater among those with obesity, affecting up to 30% of these adolescents [7]. This data are particularly worrying for the foreseeable future, because adolescents with prediabetes also have a greater risk of developing T2D.

In addition, the course of T2D that starts at a young age may be clinically more aggressive, and chronic vascular complications of diabetes appear earlier than in older individuals [5,8].

T2D is considered as the consequence of a close interaction between genetic and environmental factors; that is, environmental factors promote the expression of the phenotype of a genetic predisposition to the disease [2]. Among environmental factors, diet is considered a ‘driving force’ in T2D development [9]. In particular, two components of the diet are mainly implicated in the development of T2D: total energy intake and diet macronutrient composition. Chronic energy intake (beyond required levels) can promote fat mass increase and body composition changes, with negative implications on glucose metabolism, i.e., glucose oxidation and storage, insulin sensitivity, and insulin secretion [10]. Additionally, a sedentary lifestyle and a low physical activity level promote a positive energy imbalance; the coexisting reduced metabolic activity of skeletal muscle affects both energy requirements and substrate oxidation, with implications in nutrient balance equations [11]. Moreover, growing evidence is available on the relationship between meal composition and postprandial blood glucose profile [12,13,14]. Blood glucose levels are mainly affected by dietary carbohydrates; however, protein, fat, and fiber, especially if taken in large amounts, may also influence postprandial blood glucose levels [12,14,15]. In particular, dietary carbohydrates are thought to play a key role in T2D management [16]. Indeed, high consumption of some types of carbohydrates, such as rapidly digestible carbohydrates and, in general, carbohydrates with a high glycemic index (GI), has been associated with poor glycemic control in people with T2D, whereas high consumption of fiber has been associated with a better glycemic control [9].

The strong association between dietary carbohydrates and glycemic control demonstrated in individuals with T2D further emphasizes the need to identify for each subject the total amount of carbohydrate intake, as well as the best carbohydrate composition of his/her diet, also taking into account the protein, fat, and fiber contents of the diet. Therefore, the main aim of this narrative review is to briefly discuss the current knowledge about the role of dietary carbohydrate intake in T2D prevention and treatment, underlining how diet may be modulated to reduce risk of incident T2D and ameliorate the management of the disease.

## 2. Methods

References for this narrative Review were identified through searches of PubMed, Medline and Cochrane Library databases with the following search terms: “type 2 diabetes and nutrition”, “type 2 diabetes and carbohydrates”, “type 2 diabetes in adolescents”, “type 2 diabetes in young people”, “type 2 diabetes and gut microbiota”, “type 2 diabetes and genetics”, and “low-calorie sweeteners and glucose metabolism” from the inception date to 20 September 2021. The narrative review includes articles published exclusively in English. In total, 243 articles were initially selected from which 130 were included in the review. Since this is not a formal systematic review, we cannot present a Preferred Reporting Items for Systematic Reviews and Meta-Analyses (PRISMA) flow diagram for search and selection processes of the articles included in the review.

## 3. Nutrition and Dietary Macronutrient Composition, Genetics and Gut Microbiota

### 3.1. Carbohydrates

Carbohydrates are a group of different substances with heterogeneous chemical and physical properties. Carbohydrates are implicated in different processes, such as energy metabolism, glycemic control and insulin secretion, lipid metabolism, and colonic function [17].

The first classification of carbohydrates is based on the degree of polymerization, as proposed at the Joint Food and Agriculture Organization/World Health Organization (WHO) Expert Consultation on Carbohydrates in 1997 [18]. This chemistry-based classification divides carbohydrates into three main groups: sugars (monosaccharides, disaccharides or sugar alcohols), oligosaccharides (short-chain carbohydrates made by 3–9 monosaccharides) and polysaccharides (made by more than 9 monosaccharides) [17,18]. However, this classification is not useful to assess the nutritional effects of carbohydrates because it does not consider carbohydrate bioavailability [17].

For this reason, another classification has been proposed by the WHO that divides carbohydrates into two groups mainly depending on their digestibility: glycemic (or available) or nonglycemic (or unavailable) carbohydrates [19]. Glycemic carbohydrates are hydrolyzed to monosaccharides, which are absorbed in the small intestine, whereas nonglycemic carbohydrates are not hydrolyzed by human enzymes and may be fermented in the large intestine [18]. Carbohydrate digestion is important for the nutritional impact of carbohydrates on glucose homeostasis and for the metabolic and endocrine responses to their ingestion [20]. Glycemic carbohydrates include sugars (apart from alcohol sugars) and starch (wheat, rice, maize, potato), which offer rapidly and slowly available glucose, respectively. Nonglycemic carbohydrates include nonstarch polysaccharides (found in fruits and vegetables), resistant starch (whole grain cereals, legumes, and green bananas), resistant short-chain carbohydrates (nondigestible oligosaccharides) and alcohol sugars [18,19]. The first three types of nonglycemic carbohydrates are also known, by a more general term, as dietary fiber [21] (Table 1).

Despite this more practical classification, it is not easy to assess the nutritional effects of carbohydrates, because starch intake is often reported as total starch intake without specifying the subtype and digestibility [22]. Moreover, different terms can be used to describe the same carbohydrate. For instance, to describe sugars, many terms are used (such as sugars, total sugars, total available sugars, free sugars, added sugars, refined sugars, simple sugars, or caloric sweeteners), leading to a more confusing interpretation of their nutritional effects [17]. For the above-mentioned reasons, the actual types of carbohydrate consumed and their consequent nutritional impacts on glucose metabolism are not easy to assess [22]. As a consequence, the classification based on carbohydrate digestibility could be considered useful, but its use is not always accurate.

### 3.2. Glycemic Index and Glycemic Load

Postprandial glycemia strongly depends on the total carbohydrate content of a meal. The effect of dietary carbohydrate intake on glycemia occurs due to the conversion of carbohydrates into glucose, which in turn is the strongest insulin secretagogue [23]. Therefore, the assessment of the quantity and quality of carbohydrates in the diet is crucial for clinical purpose [24].

The glycemic index (GI) is defined as the glycemic response promoted by 50 g (or in some cases by 25 g) of a glycemic carbohydrate in a specific food, expressed as a percentage of the glycemic response promoted by 50 g (or by 25 g) of a reference carbohydrate (pure glucose or white bread) [25]. The GI is determined by evaluating the incremental area under the curve for blood glucose levels over 2 h after the ingestion of the test food containing 50 g of carbohydrate compared with the standard food (glucose or white bread), and the results are expressed as a percentage of the test food relative to the standard food [26]. In general, carbohydrates that are rapidly digested and absorbed have a high GI (GI ≥ 70 by comparison with the scale of glucose), whereas carbohydrates that are slowly digested or absorbed have a lower GI (GI ≤ 55 by comparison with the scale of glucose) (Figure 1).

The glycemic load (GL) is the result of the product between the GI and the total available carbohydrate ingested in a meal [25]. The GI suggests the glycemic impact of carbohydrates regardless of quantity, whereas the GL indicates the overall glycemic impact considering both the quality and quantity of carbohydrates [28]. Some authors have suggested that dietary GL is a better indicator of postprandial glycemia and insulin demand than carbohydrate content alone in healthy individuals [29]. However, the glycemic response after food intake is affected by many factors other than the GI and GL, since a meal is usually characterized by the ingestion of different types of food, which affect nutrient digestion and absorption. The glycemic response, in fact, is influenced not only by the rate of carbohydrate digestion but also by the amount of carbohydrates consumed, the volume and macronutrients composition of the meal, the duration of fasting before the meal, the level of physical activity, the gastrointestinal response (for example, the speed of gastric emptying), and the metabolism after absorption, which is influenced by different foods ingested [19].

Proteins, for example, may reduce the glycemic response after a meal, and several mechanisms may be implicated in the process. Some proteins, especially whey proteins, may induce insulin secretion and incretin stimulation, thereby improving postprandial glycemia. It is possible that both the incretin effect and the direct stimulation of pancreatic beta cells are involved in the process [12]. This mechanism is particularly active with dietary proteins that are rapidly absorbed, whereas is not observed, for example, after the ingestion of cod or wheat gluten (which are not rapidly absorbed) [12,30]. Moreover, whey proteins may slow gastric emptying if they are co-ingested with carbohydrates or ingested before them (as a preload) [12]. Slower gastric emptying has been attributed to protein stimulation of gut hormones (e.g., cholecystokinin and incretins) [13]. Co-ingested fats, as well as proteins, may slow the emptying of the stomach, inducing a reduced postprandial rise in plasma glucose and insulin levels [14].

Moreover, cooking may also influence the glycemic response [31]. After the ingestion of raw starch, in fact, smaller responses of plasma glucose and insulin levels occur compared to the ingestion of cooked starch. It is likely that this difference is mostly due to the reduced availability of raw starch for enzyme digestion [32]. Finally, the digestibility of a food may also be influenced by its form [33]. For instance, different forms of pasta show different GIs: macaroni pasta has a greater GI than spaghetti pasta [31].

As a consequence of all these influences, the glycemic response after a meal does not simply reflect the glycemic response predicted considering only the GI values of ingested carbohydrates [19]. In other words, the GI is only one of the factors that may affect glycemic responses after a mixed meal [34].

Some authors, in fact, suggest that dietary recommendations based on the GI may be deceptive, because there is an important intra- and inter-individual variability in glycemic response and because foods with a low GI are not always foods with a high nutritional value and vice versa (i.e., a high GI does not always mean low-quality food) [26,35]. For example, some confectionary products, such as chocolate, may have a low GI but also contain saturated fats that reduce their nutritional value [27,36]. For this reason, the GI should not be considered alone, but in conjunction with all the other components of a specific food [27].

In 2005, the American Diabetes Association (ADA) suggested that the “use of the GI and GL can provide an additional benefit over that observed when total carbohydrate is considered alone” [37], thus further underlining the importance of the quality of dietary carbohydrates [38]. However, in 2019, a consensus report of the ADA suggested that the GI and GL do not have a relevant impact on glycosylated hemoglobin (HbA1c) or fasting glucose levels in individuals with or at risk for T2D [39].

Despite controversies in the utility and feasibility of the GI and GL in dietary recommendations, some studies suggest a significant association between a low-GI diet and a lower risk of developing T2D [25]. In a study conducted on obese adolescents, a 6-month diet with a reduced GL was associated with body weight reduction and improved insulin resistance compared to a low-fat diet [40]. In another study conducted on children and adolescents with T2D or impaired glucose tolerance, the usefulness of a low-GI diet was suggested by the presence of lower postprandial blood glucose levels and lower glycemic variability [41]. In adults, a 3-month low-GI diet (with a high content of legumes or fiber) has been associated with improved glycemic control [42]. Furthermore, in another study, the intake of acarbose, i.e., an inhibitor of starch conversion into glucose (in this way acarbose mimes a low-GI diet), significantly reduced the risk of incident T2D in subjects at high risk for T2D [43]. Finally, a recent meta-analysis showed that diets with a higher GI and GL were associated with a higher risk of developing T2D, particularly if subjects also consumed a diet with a low content of dietary fiber [44].

Moreover, in addition to the negative effects on postprandial blood glucose levels, high-GI/GL meals seem also to promote excessive weight gain due to increased hunger and decreased satiety. High-GI foods, in fact, stimulate the production of higher circulating levels of insulin, which promotes cellular glucose uptake, inhibition of hepatic gluconeogenesis and suppression of lipolysis. The relative hyperinsulinemia induced by high-GI foods leads to a decrease in blood glucose levels, which could be responsible for hunger and over-eating [25].

However, despite this evidence, several authorities do not give any specific suggestions regarding the intake of low-GI/GL foods to prevent or treat T2D [45,46,47]. This likely depends on the fact that the GI and GL are not practical and feasible indexes to predict the glycemic response [36].

### 3.3. T2D and Carbohydrates: ‘Two Sides of the Same Coin’

In people with T2D, carbohydrates may have not only a detrimental but also a beneficial effect on glucose metabolism, depending on their type [15,39]. Nonglycemic carbohydrates, in fact, are associated with a lower risk of incident T2D [15]. In particular, nonstarch polysaccharides (found in fruits and vegetables), resistant starch (whole grain cereals, legumes, and green bananas) and resistant short-chain carbohydrates (nondigestible oligosaccharides), which are all known as dietary fiber, may have a beneficial effect on glucose metabolism [15]. Indeed, in a study conducted on overweight adolescents, an increase in dietary fiber was associated with a lower risk of T2D development [48]. In 109 overweight children and adolescents, an increase in dietary fiber improved metabolic health [49]. In adults, recent systematic reviews and meta-analyses reported consistent evidence that diets with higher intakes of dietary fiber or whole grains were significantly associated with a lower incidence and a reduced mortality from several non-communicable chronic diseases, such as T2D [36]. In another study, higher consumption of total whole grains (and in general of whole grain foods such as whole grain cereal, porridge, dark bread, brown rice, and bran) was associated with a lower risk of T2D [50]. Similarly, Sun et al. evaluated the association between whole grain wheat and rye and the risk of T2D or impaired glucose tolerance and found that higher consumption was associated with a lower risk of both diseases [51].

In general, the suggested intake of dietary fiber is around 10 g/1000 kcal per day. Higher intakes could provide even more benefits. Unfortunately, the majority of people worldwide consume less dietary fiber than the suggested amount [21]. Fiber may have beneficial effects on blood glucose levels through different mechanisms. Fiber decreases systemic low-grade inflammation, affects gut microbiota composition, reduces intestinal permeability and delays gastric emptying, inducing a slower entry of glucose into the blood [23].

However, other types of starch (available or glycemic starch) may have a detrimental effect on glucose metabolism [52]. A meta-analysis showed that higher intake of white rice was associated with a higher risk of incident T2D [53]. Consistently, in a study conducted on 39,765 men and 157,463 women, a higher intake of white rice was associated with a higher risk of incident T2D [54]. In the same study, it was also confirmed that a higher intake of brown rice (a more resistant starch) was associated with a lower risk of incident T2D [54]. In another study conducted on 84,555 women followed for nearly 20 years, a modest positive association between dietary intake of potatoes (a digestible starch) and the risk of incident T2D was observed, particularly when potatoes were substituted for whole grains [55].

Regarding sugars, there is evidence of an association between a high consumption of sugars and risk of incident T2D [15]. However, not all sugars have the same effect as sucrose and glucose, which promote the highest glycemic response [15]. Fructose intake promotes a lower glycemic response, and when it is consumed in its primary form (such as in fruits), it does not exert any harmful effect due to its slower digestion [56]. According to the WHO recommendations, the intake of free sugars should be less than 10% of the total energy intake. Free sugars include monosaccharides and disaccharides added to foods and beverages and sugars present in honey, syrups, fruit juices and fruit juice concentrates [57]. The detrimental effect of sugars is likely to be mediated by their capacity to promote a high glycemic response after consumption, independent of their effect on body weight [15]. In general, the consumption of glycemic carbohydrates (as sugars or rapidly digested starch) promotes a quick increase in plasma glucose levels with a consequent high secretion of insulin from pancreatic beta cells. The excessive and chronic production of insulin could lead to exhaustion of pancreatic beta cells with consequent development of T2D over time [52,58] (Figure 2).

Concerning the daily total amount of carbohydrate intake, it seems that a wide range of intakes is acceptable, underlining the importance of the quality of carbohydrates instead of the quantity, although some studies suggest that a low-carbohydrate diet may be beneficial for management of T2D [36].

In a recent review of studies involving young individuals, a low-carbohydrate diet was associated with a lower risk of T2D [59]. In another study, conducted in a group of overweight or obese adults with T2D and HbA1c > 6% (>48 mmol/mol), a very-low-carbohydrate ketogenic diet was compared to a moderate-carbohydrate, calorie-restricted, low-fat diet. Notably, after 12 months, T2D subjects who followed a very-low-carbohydrate diet had better HbA1c levels, greater weight loss and even lower need for glucose-lowering drugs [60]. However, a systematic review and meta-analysis of 23 studies suggested that low-carbohydrate diets could have just a modest, short-term beneficial effect on glycemia [61].

The ADA guidelines suggest that low-carbohydrate and very-low-carbohydrate (<25% total energy) eating patterns are associated with a significant reduction in HbA1c levels and a lower need for anti-hyperglycemic agents in individuals with established T2D [62,63]. However, an ideal suggested percentage of energy from carbohydrates is still not available. For instance, the Mediterranean diet, which is not particularly low in carbohydrates, was consistently demonstrated to be appropriate for individuals with T2D [64,65]. This finding suggests that the proportion of different macronutrients (and carbohydrates) in the diet seems to be less important than their quality [61].

Dietary carbohydrates may be also implicated in the development of non-alcoholic fatty liver disease (NAFLD), which is the most common liver disease in the general adult population in many parts of the world [66]. NAFLD and T2D are known to frequently coexist and act synergistically to increase the risk of adverse (hepatic and extra-hepatic) clinical outcomes [67]. Furthermore, convincing evidence now indicates that NAFLD is associated with a two-fold increased risk of incident T2D, independent of obesity and other common metabolic risk factors [68]. It has been demonstrated that high consumption of fructose may lead to the development of NAFLD when it is not consumed in its primary form (such as in fruits) [56,69]. Consistently, the reduction of fructose intake may improve hepatic fat content in obese youths, even without modifying total energy intake [70]. Sucrose, as well, has been associated with the development of NAFLD and metabolic syndrome features in rats [71]. Conversely, dietary fiber may exert some beneficial effects on NAFLD development, possibly throughout the reshaping of gut microbiota [72].

Collectively, therefore, different carbohydrates may exert different effects on glucose metabolism. Therefore, it is not possible to make a general suggestion on the better proportion of carbohydrates for a diet, because the effects of their ingestion appear to be more associated with the different types of carbohydrates consumed than with their quantity.

### 3.4. Genetics and Carbohydrate Metabolism: Individual Variability in Response to Dietary Interventions

Individual risk of T2D is strongly associated with genetic factors [73]. A study conducted on 343,237 Danish individuals with established T2D reported an important family aggregation of the disease due to the association with genetic predisposition [74]. Consistently, the high concordance rate for T2D (96% considering either T2D and impaired glucose tolerance) in monozygotic twins further underlines the important role of genetic factors in the development of T2D [75].

Interest in genome-wide association studies (GWAS) in the last decade has led to a deeper knowledge of T2D genetics [76]. Some GWAS studies and exome sequencing studies have identified hundreds of genomic loci associated with a higher risk of developing T2D and related metabolic traits [77,78]. The identification of these genetic variants might also explain some pathogenic mechanisms of T2D, which could be useful to improve therapeutic interventions [77].

Among therapeutic strategies, diet is considered one of the most important actions to prevent and manage T2D. However, significant inter-individual differences in response to diet interventions have been reported to be probably due to the interaction with genetic factors [79]. Despite this evidence, current dietary guidelines are mainly based on population data and often do not consider the inter-individual variability in responses to different diets [76]. Nevertheless, studies suggested that some genetic variants could modify the effects of diet on insulin resistance and body weight, which are two strong risk factors for T2D development [79]. Zheng et al., for example, showed an association between genetic variants of insulin receptor substrate 1 (IRS1) and dietary fat and carbohydrate intakes. In fact, two polymorphisms of IRS1 (rs7578326 G allele and rs2943641 T allele) showed a beneficial effect on insulin resistance and metabolic syndrome features only when a low-fat and high-carbohydrates diet (with a low GI and GL) was followed [80]. In another study, Zheng et al. assessed the genes encoding gastric inhibitory polypeptide (GIP) and its receptor (GIPR), which have been associated with insulin resistance and obesity. The authors found that a specific variant of GIPR (the T allele of GIPR rs2287019) was associated with a significant improvement in glucose homeostasis after a low-fat, high-carbohydrate (with high fiber content) diet [81]. This effect could be mediated by decreased GIP secretion due to the diet (low-fat diet) and/or by impaired GIPR function due to GIPR genetic variants, but more investigations are needed to better clarify the underlying mechanisms involved in this process [81]. Consistently, another study reported an association between a variant of the proprotein convertase subtilisin/kexin type-7 (PCSK7) rs236918 G genetic variant and the response to a specific diet. In particular, this genetic variant was associated with a significant decrease in plasma insulin levels in individuals who consumed a high-carbohydrate diet in a period of weight loss. However, the underlying mechanisms involved in this process are not entirely understood [82].

Moreover, genetic variants in the fibroblast growth factor-21 (FGF-21) gene could also influence the response to a specific diet. In detail, in the presence of the “carbohydrate intake-decreasing allele” of the FGF-21 genetic variant, it seems that a low-calorie, high-carbohydrate and low-fat diet improves abdominal obesity and that, conversely, there is a smaller reduction in abdominal obesity in response to a low-carbohydrate and high-fat diet [83].

A small case-control study (involving 159 obese individuals and 154 normal-weight controls) examined the relationship between the β2-adrenoceptor gene polymorphism Gln27Glu, which is associated with a higher risk of obesity, and the response to a high-carbohydrate diet. In the presence of obesity, there was a close interaction between this genetic variant and dietary carbohydrate intake. In fact, women carrying the β2-adrenoceptor gene polymorphism Gln27Glu with a higher carbohydrate intake had a greater risk of obesity and higher plasma insulin levels than women with the same polymorphism who consumed a lower amount of carbohydrates [84]. In another case-control study (obese subjects vs. lean controls), the relationship between a genetic variant (the Pro12Ala polymorphism) of the peroxisome proliferator-activated receptor gamma gene (PPAR gamma) and the risk of obesity depending on dietary intake was explored [85]. Among individuals who consumed a high-carbohydrate diet, there was an increased risk of obesity in the presence of this polymorphism [85]. Moreover, in the TULIP study, an association between increased weight loss and a high-fiber diet (>25 g daily) in individuals homozygous for a specific genetic variant (rs7903146) of the transcription factor 7-like two (TCF7L2) gene was demonstrated [86].

Finally, in another study conducted on 6934 women and 4423 men, the interaction between sugar-sweetened beverage intake and genetic predisposition in relation to body mass index (BMI) was analyzed. Genetic predisposition was evaluated considering 32 genetic loci that were associated with BMI. The results obtained suggested that the genetic association with high BMI values was more significant when there was a greater consumption of sugar-sweetened beverages [87].

According to all these studies, a specific dietary intervention could be more appropriate than others depending on individual genetic variants, and on the basis of genotype, a personalized nutritional approach might be selected [88]. As a consequence, particularly in individuals who are genetically predisposed to obesity and related metabolic diseases, it is important to define the carbohydrate quality and quantity of a diet [89].

### 3.5. Low-Calorie Sweeteners

The growing pandemic of obesity and T2D has led to the implementation of efforts to promote healthier eating habits and to contrast the overconsumption of added sugars [90,91]. In this context, low-calorie sweeteners (LCSs) have gained a role in maintaining a sweet taste without adding extra energy, as occurs with foods and drinks containing added (or caloric) sugars [90].

Low-calorie sweeteners, in fact, have the capability to provide sweetness by adding little or no energy [92]. Different terms are used to identify these compounds, such as “low-calorie sweeteners”, “high-intensity sweeteners”, “artificial sweeteners”, “noncaloric sweeteners”, “high-potency sweeteners”, “nonsucrose sweeteners”, “intense sweeteners”, “nonnutritive sweeteners”, or “sugar substitutes” [90,92]. Moreover, many sweeteners, such as acesulfame-K, aspartame, neotame, saccharin, sucralose, and stevia, which are natural compounds, have been approved by the Food and Drug Administration (FDA) [90].

Since FDA approval, LCS usefulness and their beneficial effects on metabolic control have been questioned. The open question is whether LCSs are truly useful and beneficial to lower energy intake and body weight and to better control swings of glycemia in people with T2D [90].

Recently, some authors have suggested that LCSs reduce energy intake and may promote weight loss when substituted for caloric sugars (such as sucrose) [91,93]. As a consequence, LCSs could be a strategy to consider to lower the prevalence of obesity and T2D [91,94]. Moreover, these authors suggested that LCSs do not adversely affect glycemic control (HbA1c, fasting glucose or postprandial glucose levels) or insulin regulation in individuals with T2D [91]. In a trial involving 641 normal-weight children, the replacement of sugar-sweetened beverages with LCS-added beverages for 18 months significantly reduced weight gain and body fat gain [95]. Moreover, a recent meta-analysis reported that replacing caloric sugar with LCSs led to weight reduction in obese and overweight adults [96]. In another meta-analysis, the consumption of LCSs was not associated with increases in plasma glucose levels among adult individuals [97].

However, there are also other studies that underline the potential negative or neutral effects of LCSs on glucose metabolism. A systematic review and meta-analysis evaluating the effects of LCSs in both adults and children concluded that there is no convincing evidence to date to recommend the use of LCSs and that harmful effects could not be excluded [98,99]. In a study conducted on 3682 adults, long-term consumption of LCSs was associated with increased body weight [100]. It seems that habitual LCS consumers uncouple sweet taste from energy intake and, as a consequence, increase food intake to reach “energy compensation” [92]. In a study conducted in children, in fact, there was complete caloric compensation during a meal 30 min after the consumption of a sucralose- versus a sugar-added drink. Therefore, the children’s intake of calories during the meal depended on the calories contained in the “preload” drink [101]. This could be a consequence of the different brain stimulation induced by LCSs and caloric sugars [92].

Recently, the use of aspartame, a low-calorie sweetener, was associated with insulin deficiency and greater insulin resistance [102]. It has been reported that aspartame use in people with T2D could be also associated with weight gain and that it might act as a chemical stressor inducing an increase in plasma cortisol levels, which are implicated in glucose metabolism [102]. Moreover, an experimental study conducted in mice found that LCSs induce glucose intolerance by altering gut microbiota [103]. LCS-consuming mice developed increased glucose intolerance compared to different controls (water-, sucrose-, or glucose-consuming mice). The observed adverse effect on glucose metabolism was abrogated by antibiotic treatment, and fecal transplantation from LCS-consuming mice to germ-free mice induced negative effects on glucose metabolism. These two results clearly suggest that gut microbiota may be involved in this process [103].

Currently, the ADA guidelines suggest that LCSs, especially if consumed in moderation, may be useful in some T2D subjects, who are used to regularly consuming sugar-added products to substitute as non-nutritive sweeteners [62]. Indeed, although LCSs do not seem to have a significant effect on glycemic control, they can reduce caloric and carbohydrate intake if the patient does not compensate with calories from other products [62]. The ADA guidelines also suggest that people with T2D should decrease their daily consumption of both sugar-added and LCS-added products [62].

To conclude, the contrasting evidence about LCSs underlines the need to enhance our current knowledge(s) to assess their usefulness for the prevention and management of both obesity and T2D. Moreover, each LCS has its own chemical structure, and for this reason, it should be considered individually to better understand its efficacy and safety [92].

### 3.6. Gut Microbiota, T2D, and Carbohydrates

In recent years, evidence has accumulated to show an association between gut microbiota and the development of several human diseases, such as T2D, although a relationship of causality is still not certain [76,104]. Gut microbiota are implicated in some physiological processes, such as energy (and, in particular, glucose) metabolism, metabolic pathways, immune responses and regulation of gut integrity and mobility [104]. In particular, the role of gut microbiota in glucose metabolism has been confirmed by recent studies. Vrieze et al. analyzing the effect of microbiota transplantation from healthy individuals to those with metabolic syndrome reported that 6 weeks after transplantation, recipients had an improvement in insulin resistance [105]. Koote et al. also reported that 6 weeks after microbiota transplantation from healthy persons to obese individuals with metabolic syndrome, there was an improvement in insulin resistance in transplant recipients and that this beneficial effect was mainly influenced by changes of gut microbiota composition [106]. This awareness has led to exploring how gut microbiota may contribute to the development of T2D and how it may be modulated to improve the glycemic control of the disease. Several studies have recently reported that gut microbiota of individuals with T2D has some differences when compared to gut microbiota of healthy people. In detail, individuals with T2D have a decreased presence of butyrate-producing bacteria in their gut microbiota [107,108,109]. Butyrate is the most important energy source for enterocytes and has a main role in maintaining the integrity of the intestinal barrier [110]. Decreased butyrate levels, in fact, could induce damage to the intestinal mucus barrier because bacteria utilize mucus glycoproteins as a nutrient source instead of butyrate [111].

Moreover, a specific bacterium that has been associated with T2D and obesity is mucus-colonizing *Akkermansia muciniphila*. Recently, Yassour et al. found that gut microbiota of individuals with obesity and insulin resistance show a decreased abundance of *A. muciniphila* [112]. Consistently, in another study, higher levels of *A. muciniphila* and greater microbial diversity were associated with a better metabolic status in overweight and obese individuals, and higher levels of *A. muciniphila* could induce a better improvement in glucose homeostasis after a calorie restriction diet [113].

Furthermore, another difference was reported in a study conducted on 277 nondiabetic Danish individuals. In this study, the authors found that in the serum metabolome of individuals with insulin resistance there were more branched-chain amino acids (BCAAs) due to an increase in *Prevotella copri* and *Bacteroides vulgatus* in their gut microbiota [114]. *P. copri* and *B. vulgatus* are considered the main producers of BCAAs and could be implicated in insulin resistance. Experimentally, it has been demonstrated that *P. copri* promotes insulin resistance and glucose intolerance and may increase serum levels of BCAAs in mice [114].

Gut microbiota of individuals with T2D are also characterized by a more proinflammatory environment. Gut microbiota homeostasis has a major role in the control of the immune response and chronic inflammation, and as a consequence, intestinal dysbiosis may affect immune cells throughout the production of lipopolysaccharides (LPS) and other metabolites [115]. Gut microbiota of individuals with T2D are characterized by a high number of opportunistic pathogens (such as *Clostridium clostridioforme*, *Clostridium hathewayi*, *Clostridium ramosum*, *Clostridium symbiosum*, *Eggerthella* sp., and gram-negative *Escherichia coli* and *Bacteroides caccae*) [108,116]. Intestinal dysbiosis (with increased intestinal permeability) leads to higher serum levels of LPS produced by gram-negative bacteria, and in turn, LPS may promote a proinflammatory response by stimulating toll-like receptors with the consequent release of multiple inflammatory cytokines [116,117]. LPS can also promote the stimulation of some pathways that are implicated in the development of insulin resistance in the liver, skeletal muscle, and adipose tissue and can promote the development of T2D [117]. Therefore, as reported in the aforementioned studies and in a study by Le Chatelier et al., gut microbiota of subjects with insulin resistance are typically characterized by lower microbial diversity, a decreased presence of butyrate-producing bacteria, and a pro-inflammatory status [109].

Diet may contribute to the composition and richness of gut microbiota, and carbohydrates, among the macronutrients, seem to have the most important role in this process [110,118,119]. The majority of food that reaches the large intestine, in fact, is represented by carbohydrates and, specifically, by indigestible polysaccharides (such as resistant starch and nonstarch polysaccharides). Monosaccharides and disaccharides could also reach the large intestine if they are overconsumed or if they are not well digested in the upper part of the intestinal tract. Monosaccharides and disaccharides may be fermented in the large intestine and have a detrimental effect on gut microbiota, thereby promoting metabolic dysfunction [118,120,121]. Finally, some endogenous substrates (such as mucopolysaccharides) may reach the large intestine and contribute to influencing the gut microbiota [118,121]. Carbohydrates reaching the large intestine may shape the diversity of gut microbiota, selecting the growth of some bacteria instead of others [118].

Several studies on humans have repeatedly shown that indigestible carbohydrate intake is associated with increased gastrointestinal microbial richness and diversity and with metabolic and immunological improvements [122,123]. For example, in a study conducted on children from Burkina Faso, who consume more fiber than European children, their gut microbiota was characterized by a higher microbial diversity [124]. Moreover, these children showed an increased *Bacteroidetes/Firmicutes* ratio in their gut microbiota, as well as increased production of short-chain fatty acids, which, in turn, may modulate some physiological processes, such as the metabolism of carbohydrates and proinflammatory status [120,124].

In another study, after a 4-week diet with whole grain barley and brown rice, there was an important effect on fecal microbiota with a reduction in plasma interleukin-6 and C-reactive protein levels and an improvement in glucose metabolism [123]. In contrast, when a low-fiber diet is followed, proteins are also utilized from the gut microbiota to obtain energy throughout fermentation, and this process could lead to the production of some potentially toxic products (such as ammonia, phenols, and branched-chain fatty acids) [125,126]. Therefore, diet composition may be implicated in glucose metabolism even throughout the reshaping of gut microbiota.

## 4. Conclusions

In light of the increasing global prevalence of T2D and its considerable human and economic costs, the importance of the prevention and treatment of this burdensome disease is undeniable [3]. According to the ADA recommendations, prevention of T2D should be based mainly on lifestyle and behavioral interventions [11]. In this context, diet and physical activity play a major role in fighting obesity and T2D pandemics [11].

With regard to diet, current evidence suggests that there is not an ideal percentage of calories from different macronutrients (such as carbohydrates, protein, and fat) that is considered optimal for all individuals to prevent or treat T2D and that macronutrient distribution should be suggested considering the eating patterns, preferences, and metabolic goals of each person, following an individualized approach [11]. On the basis of the strong relationship between meal composition and post-prandial glucose profile, adequate attention should be paid to the mix of macronutrients ingested more than on the carbohydrate intake only. Moreover, recent data on the role of genetics and gut microbiota composition and activity contribute to increase the accuracy of personalized approaches [76,79,104] (Table 2).

With regard to physical activity, consistent evidence emphasized the crucial role of physical activity in modulating insulin sensitivity and post-prandial glucose metabolism. Indeed, 150 min/week of moderate intensity is considered beneficial in adult individuals with prediabetes, improving insulin sensitivity [11,127,128]. In adolescents and youth, daily physical activity should be always encouraged (60 min of moderate to vigorous activity) [8].

Collectively, therefore, the future of T2D prevention and treatment is a personalized nutrition, which offers the best dietary suggestions and lifestyle modifications considering the individual features of each subject [129]. Genetics plays an important role in this context due to the fact that some genetic variants may influence the metabolism of ingested foods and, as a consequence, may modulate the effects of diet on insulin resistance and body weight [79]. For this reason, in the future, genetics could have a practical application to suggest dietary choices and to help the fight against T2D pandemic [130].

## Figures and Tables

**Figure 1 nutrients-13-03344-f001:**
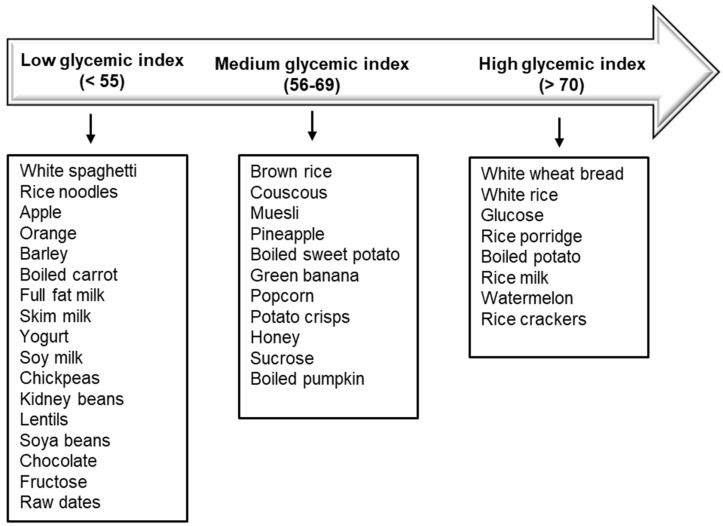
Glycemic index of different foods [27].

**Figure 2 nutrients-13-03344-f002:**
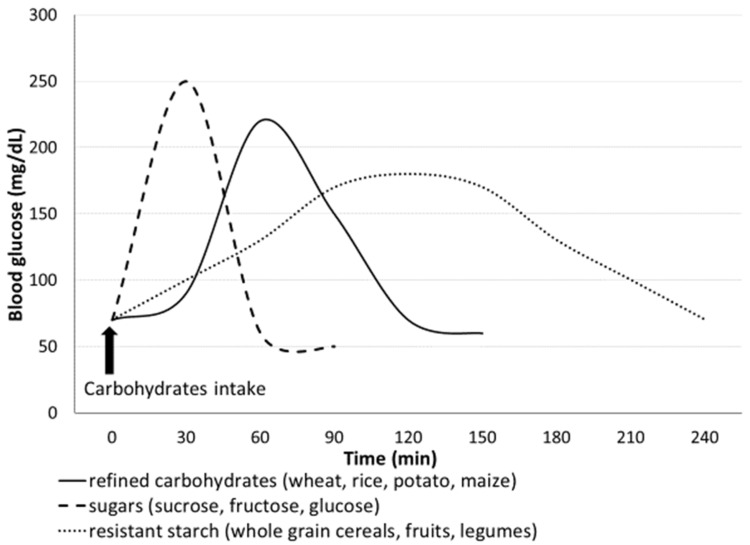
Glycemic response after intake of different types of dietary carbohydrate.

**Table 1 nutrients-13-03344-t001:** Different types of carbohydrates.

Glycemic (or Available) Carbohydrates	Non-Glycemic (or Unavailable) Carbohydrates
Sugars (sucrose, glucose, fructose)	Alcohol sugars
	Non starch polysaccharides * (in fruits and vegetables)
Starch (wheat, rice, maize, potato)	Resistant starch * (whole grain cereals, legumes, green bananas)
	Resistant short-chain carbohydrates * (non-digestible oligosaccharides)

*: fiber.

**Table 2 nutrients-13-03344-t002:** Factors involved in glucose metabolism and their effects.

Harmful Effect/Higher Glycemic Response	Beneficial Effect/Lower Glycemic Response	Variable Effect
Carbohydrates: glycemic starch (refined carbohydrates)	Carbohydrates: non-glycemic starch (fiber)	Genetics
High glycemic index foods	Low glycemic index foods	Foods co-ingested with carbohydrates (proteins, fats)
High glycemic load eating patterns	Low/very low carbohydrate eating patterns	Food form
“Overcooked” starch	“Raw” starch	
Intestinal dysbiosis	Gut microbiota richness and diversity

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
