# Peer review of "Type 2 Diabetes and Dietary Carbohydrate Intake of Adolescents and Young Adults: What Is the Impact of Different Choices?"

_nutrients, 2021, doi:10.3390/nu13103344_

Round 1
Reviewer 1 Report
The presented review article summarizes the studies necessary to understand the effect on the consumption of different types of carbohydrates and the development of type 2 diabetes disease.
The article describes the two most modern topics related to diet and diseases such as diabetes: genetics and microbiota in a detailed way and emphasizing the importance of these two factors and their relationship with carbohydrates.
The authors use an appropriate and updated bibliography through scientific articles published in recognized scientific journals.
As a personal comment, I suggest improving the presentation of the tables, especially the use of upper and lower case letters.
Author Response
RESPONSES OF THE AUTHORS TO THE REVIEWERS’ COMMENTS
Reviewer #1 (Comments to the Author):
The presented review article summarizes the studies necessary to understand the effect on the consumption of different types of carbohydrates and the development of type 2 diabetes disease.
The article describes the two most modern topics related to diet and diseases such as diabetes: genetics and microbiota in a detailed way and emphasizing the importance of these two factors and their relationship with carbohydrates.
The authors use an appropriate and updated bibliography through scientific articles published in recognized scientific journals.
As a personal comment, I suggest improving the presentation of the tables, especially the use of upper and lower case letters.
AUTHORS REPLY:
We would like to thank the Reviewer for the appreciation of our manuscript and for her/his suggestion. We have now improved the presentation of the tables.
Reviewer 2 Report
Thank you to the editor and the authors for the opportunity to review this manuscript: Type 2 diabetes and dietary carbohydrate intake of adolescents and young adults: What is the impact of different choices? The authors examine recent advances in the role of dietary carbohydrates in the prevention and management of type 2 diabetes mellitus in adolescents and young adults.
The only aspect that I think could be addressed is the methodology used. Although a narrative review does not require a formal explanation of all methodological aspects, I do think it is necessary to specify at least the inclusion and exclusion criteria as well as the flowchart.
Author Response
Reviewer #2 (Comments to the Author):
Thank you to the editor and the authors for the opportunity to review this manuscript: Type 2 diabetes and dietary carbohydrate intake of adolescents and young adults: What is the impact of different choices? The authors examine recent advances in the role of dietary carbohydrates in the prevention and management of type 2 diabetes mellitus in adolescents and young adults.
The only aspect that I think could be addressed is the methodology used. Although a narrative review does not require a formal explanation of all methodological aspects, I do think it is necessary to specify at least the inclusion and exclusion criteria as well as the flowchart.
AUTHORS REPLY:
We thank the Reviewer for this comment. However, since our article is not a formal systematic review, we could not provide a Preferred Reporting Items for Systematic Reviews and Meta-Analyses (PRISMA) flow diagram for search and selection processes of the articles included in the review. We have now added this specification in the “Methods” section in order to explain the reason why we did not present the PRISMA flow diagram.
Reviewer 3 Report
This narrative review addresses the recent advances in the role of different types of dietary carbohydrates in the prevention and management of type 2 diabetes mellitus. This is an interesting review and the authors have collected a wide dataset. The paper is generally well written and structured. However, in my opinion the paper has some minor shortcomings that should be addressed.
Image quality of Table 1, Table 2, Figure 1 and Figure 2 should be improved.
Table 1. I found this figure slightly unattractive. Text should be centered to make it more structured and the use of hyphens would not be indicated in this table.
Figure 1. I found this figure unattractive as well. Hyphens could be removed, and it would be necessary to add some table borders to frame the information.
Figure 2. Remove Time (minutes) and write Time (min) instead in X axis.
Table 2. Text should be centered to make it more structured and table borders could be modified to make this table more attractive.
Type 2 diabetes is found in scientific literature as “type 2 diabetes mellitus (T2DM)” and “type 2 diabetes (T2D)”, being the latter more common. Why did the authors use “type 2 diabetes mellitus (T2DM)” instead of “type 2 diabetes (T2D)”?
The increased prevalence of pediatric overweight and obesity has led to an augmented incidence and prevalence of type 2 diabetes mellitus, as well as the intermediate condition of prediabetes, in youth. Prediabetes should be briefly addressed within the text, especially when epidemiological data are provided. Some new citations will help the authors to provide better and more accurate information regarding prediabetes.
Orthography:
Line 86 and 87: ‘formed’, look for another synonym more appropriated (such as “made”, “composed”, etc.)
Line 112: ‘abovementioned’. Replace it by above-mentioned or aforementioned.
Line 506: remove ‘undigestible’ and write’ indigestible’ instead.
Author Response
We would like to thank the Reviewer for the appreciation of our manuscript and for her/his useful suggestions.
- All figures and tables have been improved as suggested to make them clearer and more attractive for the readers.
- We used the abbreviation “T2D” instead of “T2DM” throughout the text as suggested.
- All typos have been corrected.
- We added some information about prediabetes in youth in the “Introduction” section.
The section has been amplified as follows:
“Similarly, the condition of prediabetes in youth has risen steeply worldwide in parallel with rising rates of obesity [1]. In the National Health and Nutrition Examination Survey cohort database, 13.1% of United States adolescents had impaired fasting glycemia and 3.4% had impaired glucose tolerance after oral glucose load. In addition the prevalence of prediabetes was even greater among those with obesity, affecting up to 30% of these adolescents [2]. This data is particularly worrying for the foreseeable future, because adolescents with prediabetes also have a greater risk of developing T2D.”
- Saleh, M.; Kim, J. Y.; March, C.; Gebara, N.; Arslanian, S. Youth Prediabetes and Type 2 Diabetes: Risk Factors and Prevalence of Dysglycaemia. Pediatr. Obes. 2021, No. March, 1–8. https://doi.org/10.1111/ijpo.12841.
- Magge, S. N.; Silverstein, J.; Elder, D.; Nadeau, K.; Hannon, T. S. Evaluation and Treatment of Prediabetes in Youth. J Pediatr 2020, 219, 11–22. https://doi.org/10.1016/j.jpeds.2019.12.061.Evaluation.
- We improved the orthography as you suggested. Moreover, we provided a language revision in order to make our manuscript easier to understand and to improve the clarity.